# Recipient pre-existing chronic hypotension is associated with delayed graft function and inferior graft survival in kidney transplantation from elderly donors

**Caterina Dolla[1]☯, Alberto Mella[1]☯, Giacinta Vigilante[1]☯, Fabrizio Fop[1], Anna Allesina[1], Roberto Presta[1], Aldo Verri[2], Paolo Gontero[3], Fabio Gobbi[4], Roberto Balagna[4], Roberta Giraudi[1], Luigi Biancone[1]***

**1** Renal Transplant Center "A. Vercellone," Nephrology, Dialysis, and Renal Transplant Division, Department of Medical Sciences, "AOU Città Della Salute e Della Scienza di Torino" University Hospital, University of Turin, Turin, Italy, **2** Department of Vascular Surgery, "AOU Città Della Salute e Della Scienza" Hospital, University of Turin, Turin, Italy, **3** Department of Urology, "AOU Città della Salute e della Scienza" Hospital, University of Turin, Turin, Italy, **4** Department of Anesthesia, Intensive Care and Emergency, "AOU Città Della Salute e Della Scienza" Hospital, University of Turin, Turin, Italy

☯ These authors contributed equally to this work.
* luigi.biancone@unito.it

**Data Availability Statement:** All relevant data are within the manuscript and its Supporting information files.

## Abstract

### Background

Pre-existing chronic hypotension affects a percentage of kidney transplanted patients (KTs). Although a relationship with delayed graft function (DGF) has been hypothesized, available data are still scarce and inconclusive.

### Methods

A monocentric retrospective observational study was performed on 1127 consecutive KTs from brain death donors over 11 years (2003–2013), classified according to their pre-transplant Mean Blood Pressure (MBP) as hypotensive (MBP < 80 mmHg) or normal-hypertensive (MBP $\geq$ 80 mmHg, with or without effective antihypertensive therapy).

### Results

Univariate analysis showed that a pre-existing hypotension is associated to DGF occurrence (p<0.01; OR for KTs with MBP < 80 mmHg, 4.5; 95% confidence interval [CI], 2.7 to 7.5). Chronic hypotension remained a major predictive factor for DGF development in the logistic regression model adjusted for all DGF determinants. Adjunctive evaluations on paired grafts performed in two different recipients (one hypotensive and the other one normal-hypertensive) confirmed this assumption. Although graft survival was only associated with DGF but not with chronic hypotension in the overall population, stratification according to donor age revealed that death-censored graft survival was significantly lower in hypotensive patients who received a KT from >50 years old donor.

**Funding:** The authors received no specific funding for this work.

**Competing interests:** The authors have declared that no competing interests exist.

## Conclusions

Our findings suggest that pre-existing recipient hypotension, and the subsequent hypotension-related DGF, could be considered a significant detrimental factor, especially when elderly donors are involved in the transplant procedure.

## Introduction

Kidney transplantation is the best option for end-stage kidney disease (ESKD) treatment, improving both patients' survival and quality of life. Even though chronic hypotension significantly affects ESKD patients, few studies limited to specific time-point (i.e., intraoperative hypotension) or subsets (non-heart-beating/young living donors) [1–6] are focused on its potential role in influencing graft outcomes.

At the same time, the definition of hypotension is a matter of debate: for example, low blood pressure without a documented causative factor ("constitutional hypotension") is a disputed diagnostic entity [7]. Although literature studies generally evaluated mean blood pressure (MBP), no threshold value is universally accepted (ranging from 70 to 90 mmHg), with no consensus on whether this cut-off may vary according to age/gender [7, 8].

However, some reports suggest a higher risk of delayed graft function (DGF) secondary to a hypotension-mediated reduction in renal blood flow [9–11]. DGF, as well-known, negatively affects graft outcomes, including increased short- and long-term morbidity and mortality, allograft immunogenicity, acute rejection, hospitalization rates, and costs [1, 12–14].

Our study aims to investigate the effect of pre-existing recipient hypotension on both DGF and long-term allograft outcomes in kidney transplanted patients (KTs).

## Materials and methods

### Study design

We performed a retrospective study on 1127 patients who received their KT from brain death donors (DBD) at the *Renal Transplant Center "A. Vercellone," AOU Città Della Salute e Della Scienza Hospital, Turin* thorough an observational period of 11 years (from January 2003 to December 2013).

For every patient, blood pressure (BP) was measured using automatically inflated cuffs using a digital monitor; BP evaluation was made by health workers several times for every patient to ensure reliability and avoid values altered by emotionalism. We also considered recent BP data from their dialytic records and measurement in their waiting list schedule for patients not on antihypertensive treatment.

Based on the concordance of current values and clinical diary of the last three pre-transplant months, mean blood pressure (MBP) was calculated using the formula: MBP = [(2*DBP) + SBP]/3.

Because no guideline defines the exact cut-off value in KTs, we set the hypotension threshold at 80 mmHg according to available experiences [4, 9, 10, 15].

Thereafter, patients were divided in hypotensive (MBP < 80 mmHg; n = 65, 5.8%), normotensive (MBP ≥ 80; n = 127, 11.2%) and hypertensive group (MBP ≥ 80 with at least one antihypertensive medication, n = 935, 83%).

Before KT, all patients were on follow-up in their dialysis centers for a minimum of one year; filed medical documentation is available for every patient.

The recipients' follow-up was performed with scheduled clinical visits or hospital admissions when significant complications occurred. Data were collected from the patients' charts.

Furthermore, we evaluated donor age, gender, cardiovascular cause of death, hypertension, creatinine, KDPI and estimated glomerular filtration rate (eGFR, CKD-EPI formula) at time of organ procurement, cold ischemia time (CIT), recipient characteristics (age, gender, type of dialysis, hypertension, time spent on dialysis), and available perioperative MBP values at specific time-points (before, at reperfusion, and after surgery).

DGF was defined as the need for hemodialysis (established on a clinical basis) within the first post-transplant week.

The same group of trained pathologists analyzed all available kidney biopsies throughout the study period.

The study was performed in adherence to the last version of the Helsinki Declaration and the Principles of the Declaration of Istanbul on Organ Trafficking and Transplant Tourism. None of the transplant donors was from a vulnerable population; according to Italy legal system (*Law no. 91/1999 and Ministry of Health Decrees 8/04/2000-11/4/2008)*, all adult citizens are offered the possibility (not the obligation) of giving or withdrawing their consent to the donation of organs and tissues after death. In the case of a potential donor (i.e., a person who has been pronounced dead), the re-suscitation team will verify whether the person carries a document containing a statement of intent, or whether such a statement been registered in the online database. If a citizen has not stated his/her intention while alive, the law allows the donor next of kin to withdraw consent to organ removal while death is being ascertained.

All KT recipients signed informed consent, including their permission to have data from their medical records used in research. This study is covered by our Ethical Committee (*Comitato Etico Interaziendale A.O.U. Città Della Salute e Della Scienza di Torino—A.O. Ordine Mauriziano—A.S.L. Città di Torino*) approval, resolution number 1449/2019 on 11/08/2019 ("TGT observational study").

## Immunosuppressive regimen and perioperative management

Immunosuppressive protocols were homogeneous for the whole study period: induction was mainly conducted with anti-CD25 and steroids; calcineurin inhibitor, mycophenolate mofetil, and steroids were mostly adopted as maintenance immunosuppression. In patients who received KT from extended criteria donors (ECD), calcineurin inhibitors were introduced after functional graft recovery (usually at serum creatinine $\leq$ 2,5 mg/dL). Steroid dosage was 500 mg, 200 mg, 50 mg, and 20 mg, respectively, on days 1, 2, 3, 4 after KT, with subsequent tapering in 45–60 days.

Management of suboptimal blood pressure during intervention was also similar in the overall population and composed by initial crystalloid infusions with subsequent dopamine adoption (usually 2.5 μg/kg/min) in persistent MBP<80 mmHg.

## Statistical analysis

Discrete data were described as percentages and analyzed with the Pearson chi-square test or, for small samples, the Fisher exact test. The odds ratios (OR [CI 95%]) were used to measure relative risk. Continuous variables were described as mean ± standard deviation, when normal, and median (25th; 75th percentile), when non-normally distributed. Mann-Whitney, Kruskal-Wallis, or t-test was used when appropriate. Cumulative graft and patient survival were analyzed by Kaplan-Meier (KM) curves and the Log Rank test. Cox analysis was used for the multivariate model. Relevant variables were checked first in univariate analysis and then including multivariate analysis (logistic regression); for some continuous variables, the cut-off value was

set based on ROC curve results. The paired kidney analysis accounts for couples (n = 18) of recipients at their first KT transplanted in our center (one hypotensive and one normal/hypertensive). All statistical analyses were performed using SPSS software (IBM SPSS Statistics, vers. 25.0.0). A significant level for all tests was set at *p<0,05*.

## Results

### Characteristics of the studied population and the role of DGF

Table 1 detailed the clinical and demographic characteristics of our population.

Briefly, DGF was observed in 310 patients (28.5%); in these KTs, KM curves showed significant (p<0.001) low allograft (at 5 years 71.3% in DGF group vs. 87.9% in no DGF), patient (84% vs. 94.8%) and death-censored graft survival (80.4% vs. 91.6%) (Fig 1).

As expected, KTs who experienced DGF also had a worse renal function at any follow-up time (Fig 2).

Statistically significant associations with increased DGF risk were noted by univariate analysis for some recipient (age, history of previous KT $\geq$ 1, hemodialysis) and donor characteristics (age, death for a cardiovascular cause, hypertension, eGFR, CIT) (p = 0.01 for all these variables). As shown in Table 2, the no-DGF group was composed of younger recipients who received KT from younger donors, often without hypertension or death for a cardiovascular cause. Donor eGFR is consequently higher in the no-DGF group and, likewise, CIT is lower than in KTs who experienced DGF.

Interestingly, a close and significant correlation between pre-existing hypotension and DGF was already observed.

**Table 1. Clinical and demographic characteristics of the studied population.**

| Characteristics | All patients (n = 1127) |
|---|---|
| Recipient gender M/F, n (%) | 721 (64) / 406 (36) |
| Recipient age, yrs | 53.8 ± 11,9 |
| Recipient hypertension, n (%) | 934 (82.9) |
| HD, n (%) | 880 (78.1) |
| PD, n (%) | 247 (21.9) |
| Time spent on dialysis, yrs | 3.9 (2.7–6.7) |
| Type of transplantation (SKT/DKT), n (%) | 1082 (96) / 45 (4) |
| Previous KT (0 / $\geq$ 1), n (%) | 972 (86.2) / 155 (13.8) |
| Donor gender M/F, n (%) | 578 (51.3) / 549 (48.7) |
| Donor age, yrs | 57.3 ± 15.6 |
| Donor hypertension, n (%) | 544 (48.3) |
| Donor eGFR (CKD-EPI), mL/min/1.73m$^2$ | 89 ± 28 |
| Donor death for cardiovascular cause, n (%) | 811 (72.0) |
| KDPI≥85%, n (%) | 545 (48.3) |
| CIT, h | 17.0 [14.0–20.3]* |
| DGF, n (%) | 310 (28.5)** |

* median [25°-75° percentile]

** evaluated on available data (n = 1088)

sCr: Creatinine; SKT: Single Kidney Transplantation; DKT: Dual Kidney Transplantation; HD: Hemodialysis; PD: Peritoneal Dialysis; DGF: Delayed Graft Function; CIT: cold ischemia time

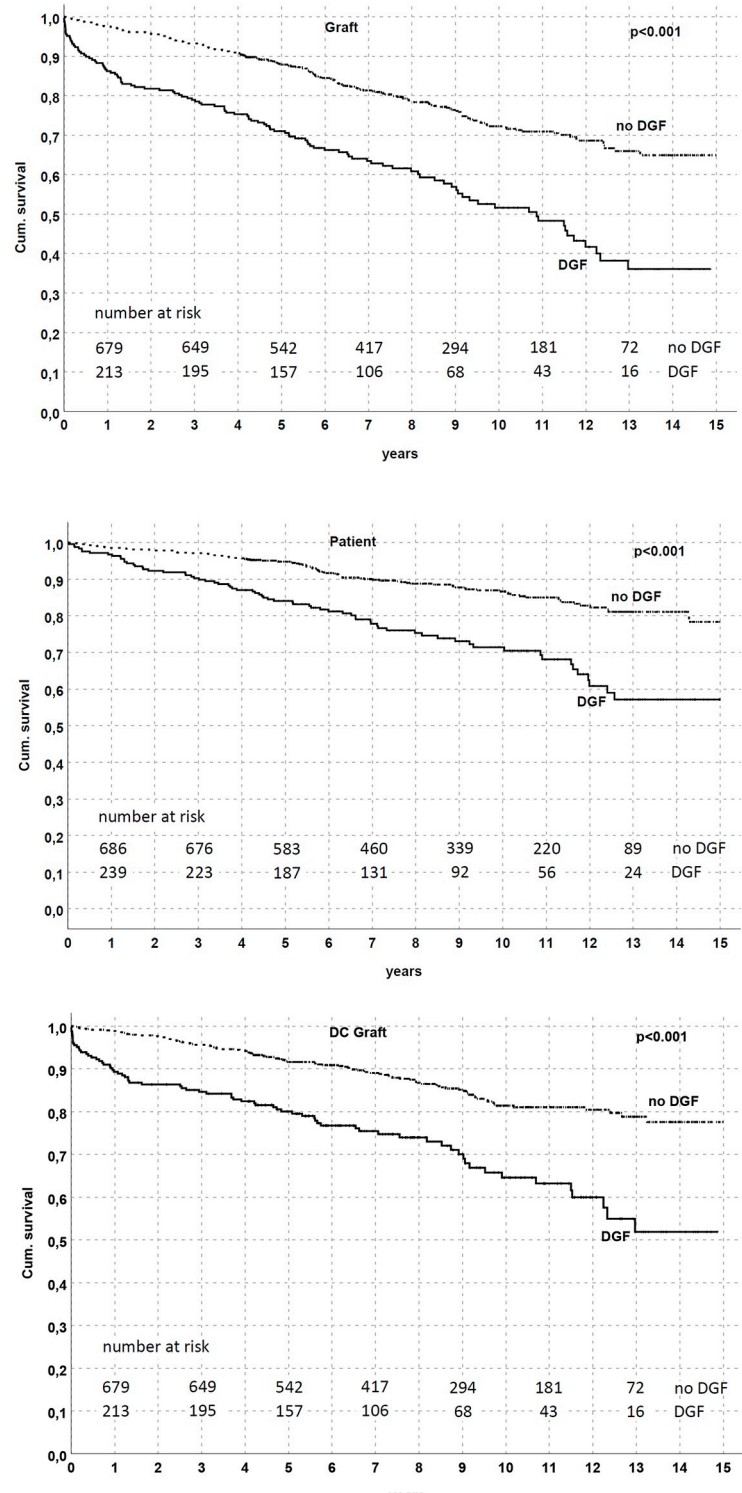

**Fig 1. Graft and patient survival by DGF in the studied population.** Graft, patient, and death-censored graft survival were all reduced in patients who experienced DGF (p<0.001 in all analyses).

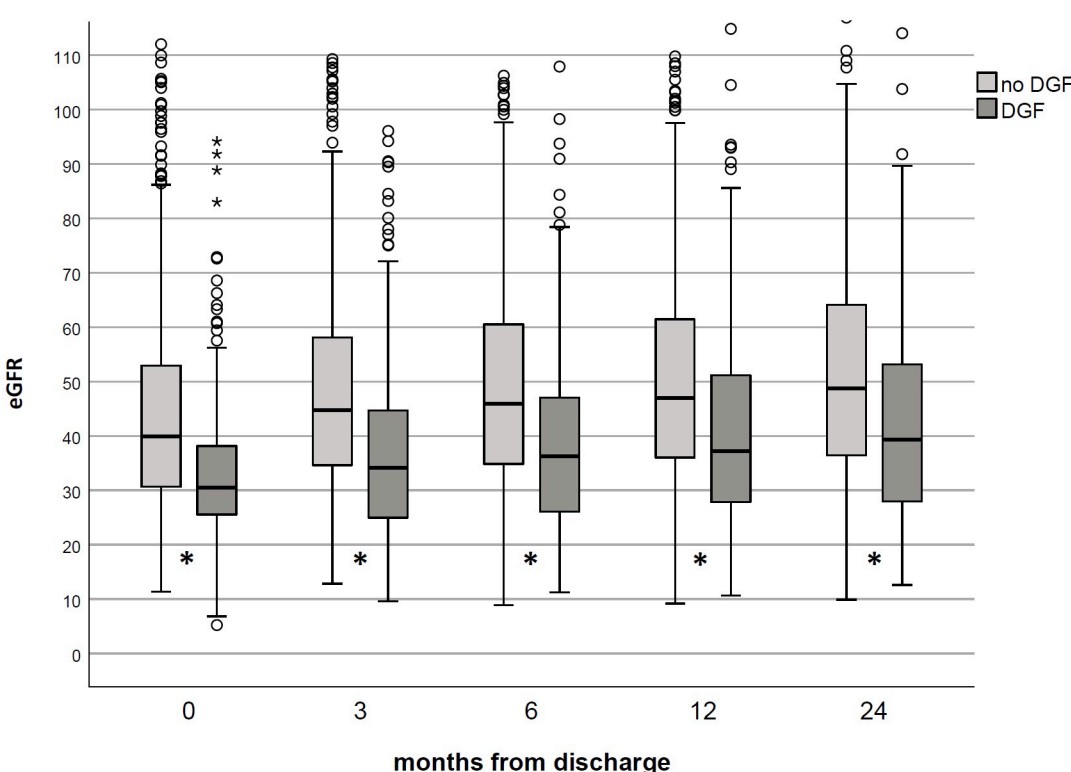

**Fig 2. Renal function in studied population according to DGF occurrence.** KTs who experienced DGF had a low eGFR (CKD-EPI formula) vs. no DGF patients at any time-point. *p<0.001.

**Table 2. Factors associated with DGF (univariate analysis).**

|  | no DGF (71.5%) | DGF (28.5%) | p |
|---|---|---|---|
| Recipient gender (M/F), % | 64,4 / 35,6 | 61 / 39 | 0.3 |
| Recipient age, yrs | 54 (45–62) | 57 (49–75) | 0.01 |
| MBP <80 mmHg, % | 3,2 / 96,8 | 12,9 / 87,1 | <0.001 |
| PD / HD, % | 25.9 / 74.1 | 10.8 / 89.2 | <0.001 [OR for HD, 2.9 (1.9–4.4)] |
| Time spent on dialysis, yrs | 3.3 (1.9–5.8) | 5.4 (3.1–7.9) | <0.001 |
| Previous KT (0 / ≥ 1), % | 89,5 / 10,5 | 79,7 / 20,3 | < 0.001 [OR, 2.2 (1.5–3.1)] |
| Donor gender (M/F), % | 50,1 / 49,9 | 53,9 /46,1 | 0,283 |
| Donor age, yrs | 58 (46–69) | 63,5 (50–72) | <0.001 |
| Donor cause of death (cardiovascular/trauma/ other), % | 67,7 / 24,3 / 8 | 81,4 / 13,1 / 5,5 | <0.001 |
| Donor hypertension (no/yes), % | 55,9 / 44,1 | 43,5 / 56,5 | < 0.001 [OR, 1.6 (1.2–2.2)] |
| Donor eGFR, mL/min/1.73m$^2$ | 93,62 (72,90–110,81) | 88,47 (64,26–104,05) | <0.001 |
| CIT, h | 16.8 (14.0–20.0) | 17.2 (14.7–20.9) | 0.01 |

MBP: Mean Blood Pressure; HD: Hemodialysis; PD: Peritoneal Dialysis; eGFR: estimated glomerular filtration rate; CIT: cold ischemia time

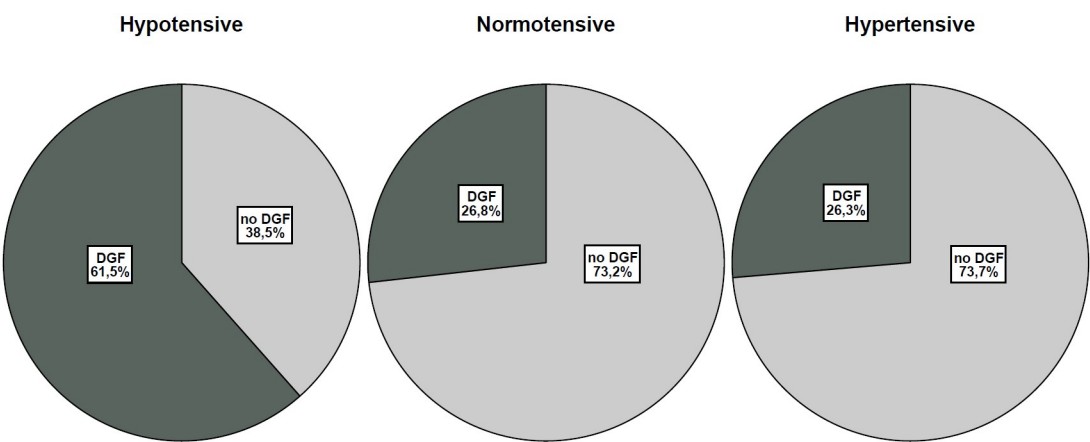

**Fig 3. DGF incidence in the hypotensive, normotensive, and hypertensive group.** DGF incidence was higher in the hypotensive group vs. normotensive and hypertensive groups (p<0.001).

## Association between pre-existing hypotension and DGF

As shown in Fig 3, DGF incidence was increased in the hypotensive vs. normal-hypertensive group (p<0.001; OR for Hypotension, 4.5; 95% confidence interval [CI], 2.7 to 7.5). This difference remained significant when separating normotensive and hypertensive KTs (Table 3), stressing the importance of a hypotension causative role in DGF occurrence.

To confirm the role of hypotension in our population and to reduce the effect of possible confounders variables, we stratified KT characteristics according to blood pressure status (Table 4); groups were quite superimposable except for reduced donor age, longer time spent on dialysis, and a higher percentage of the female gender and previous hemodialytic patients in hypotensive KTs.

The significant association between DGF and blood pressure status was confirmed by multivariate analysis including all factors associated with DGF occurrence (Table 5): patients in the hypotensive group had the higher risk of experiencing DGF (OR, 5.4; 95% confidence interval [CI], 2.8 to 10.6).

## Hypotension-related DGF may influence death-censored graft survival in KT from elderly donors

Based on the demonstration of an association between pre-existing hypotension and DGF, we consequently investigated the possible long-term effects of blood pressure status on graft and patient survival.

In the KM model, graft, patient, and death-censored survival were similar between groups (Fig 4); similarly, eGFR did not differ between hypotensive and normal-hypertensive KT (Fig 5).

**Table 3. DGF incidence according to pre-existing blood pressure status.**

|  | no DGF (71.5%) | DGF (28.5%) | P |
|---|---|---|---|
| Hypotensive group | 25/65 (38,5%) | 40/65 (61,5%) | < 0.01 |
| Normotensive group | 93/127 (73,2%) | 34/127 (26,8%) |  |
| Hypertensive group | 660/896 (73,7%) | 236/896 (26,3%) |  |

**Table 4. Characteristics of the studied population according to pre-existing blood pressure status.**

| | Hypotensive (5.9%) | Normal-hypertensive (94.1%) | p |
|---|---|---|---|
| Recipient gender (M/F), % | 47 / 53 | 65 / 35 | 0.005 |
| Recipient Age, yrs | 51.5 (42–62.25) | 55 (46–63) | 0.207 |
| PD/HD, % | 5.7 / 94.3 | 23 / 77.0 | 0.001 |
| Previous KT (0/≥1), % | 75.8 / 24.2 | 86.9 / 13.1 | 0.016 |
| Donor gender (M/F), % | 51.5 / 48.5 | 51.3 / 48.7 | 0.999 |
| Donor age, yrs | 48 (39–64.25) | 60 (48–70) | <0.001 |
| Donor cause of death (cardiovascular/trauma/other), % | 68.3 / 25 / 6.7 | 72.2 / 20.6 / 7.2 | 0.688 |
| Donor hypertension (no/yes), % | 56.1 / 43.9 | 51.4 / 48.6 | 0.525 |
| Donor eGFR, ml/min/1.73m$^2$ | 96.0 (73.8–117.2) | 91.8 (69.9–107.7) | 0.130 |
| CIT, h | 16 (13.5–19.5) | 17 (14–20.3) | 0.233 |
| Time spent on dialysis, yrs | 6.0 (3.4–8.7) | 3.8 (2.1–6.6) | <0.001 |

HD: Hemodialysis; PD: Peritoneal Dialysis; eGFR: estimated glomerular filtration rate; CIT: cold ischemia time

Nevertheless, an association between hypotension and death-censored graft survival emerged considering KTs from elderly donors. Through a stratified analysis by exploiting donor age tertiles (≤50, 51–66 and ≥67 yrs), we found that death-censored graft survival was significantly reduced (p = 0.04) in the hypotensive patients who received KT from a donor > 50 yrs (78% of functioning graft in the hypotensive group at 5 yrs vs. 87.4% in the normal-hypertensive group for donor age 51–66; 58.3% vs. 82% for donor age ≥ 67 yrs, respectively) (Fig 6).

Multivariate Cox regression analysis demonstrated that DGF and donor age were the only independent risk factors for death-censored graft survival reduction, suggesting the role of hypotension-induced DGF in this scenario (Table 6).

## Pre-existing hypotension is both associated with DGF in paired kidneys and with perioperative hypotension

We also performed a confirmatory analysis investigating KTs among paired kidneys with one hypotensive and one normal-hypertensive recipient, identifying 18 couples treated with the same induction therapy (anti-CD25 antibody and steroids): as for the overall population, DGF

**Table 5. Factors associated with DGF (multivariate analysis).**

| | P | H.R. | 95% C.I. per EXP(B) | |
|---|---|---|---|---|
| | | | Inferior | Superior |
| MBP<80 mmHg | <0.001 | 5.4 | 2.8 | 10.6 |
| Donor death for a cardiovascular cause, n (%) | 0.002 | 1.9 | 1.3 | 2.8 |
| Donor age (>60 yrs) | 0.008 | 1.7 | 1.1 | 2.4 |
| Time spent on dialysis (≥ 4 yrs) | <0.001 | 2.1 | 1.5 | 2.9 |
| CIT | 0.889 | 1 | 1 | 1 |
| Previous KT (≥ 1) | 0.078 | 1.2 | 0.9 | 2.4 |
| Recipient age (≥ 50 yrs) | 0.220 | 1.3 | 0.9 | 1.9 |
| Dialysis type (HD vs. PD) | <0.001 | 2.9 | 1.8 | 4.8 |
| Donor hypertension | 0.195 | 1.3 | 0.9 | 1.8 |

MBP: Mean blood pressure; CIT: cold ischemia time; KT: kidney transplant

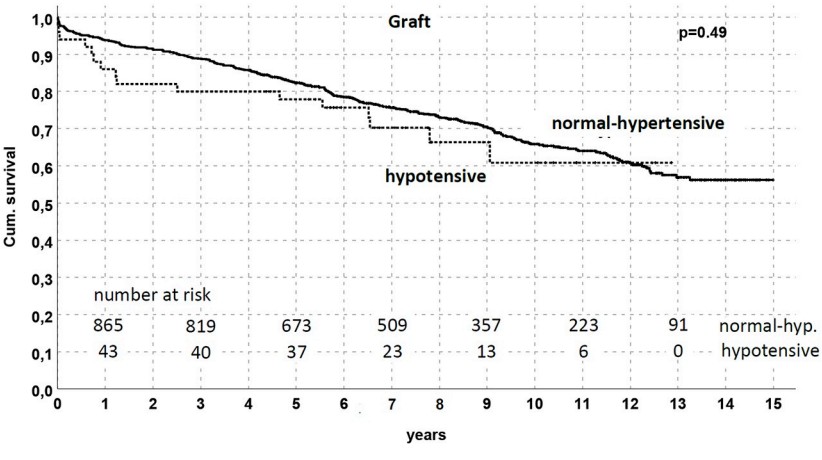

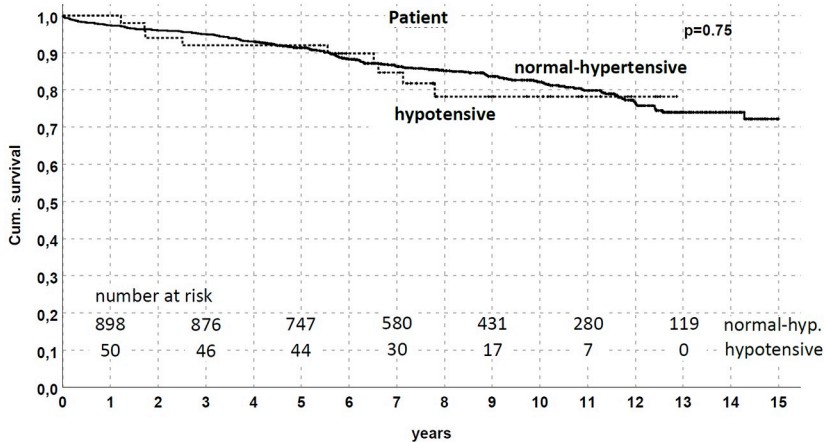

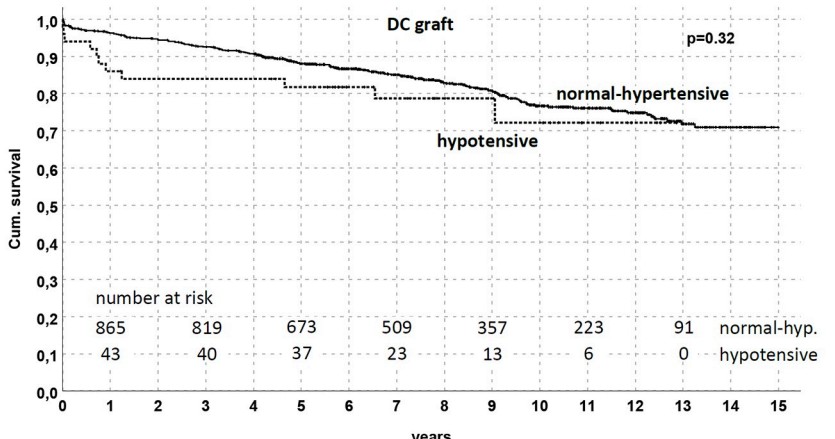

**Fig 4. Graft and patient survival by pre-existing blood pressure status in the studied population.** No difference was noted between the hypotensive and normal-hypertensive groups (p = NS in all analyses).

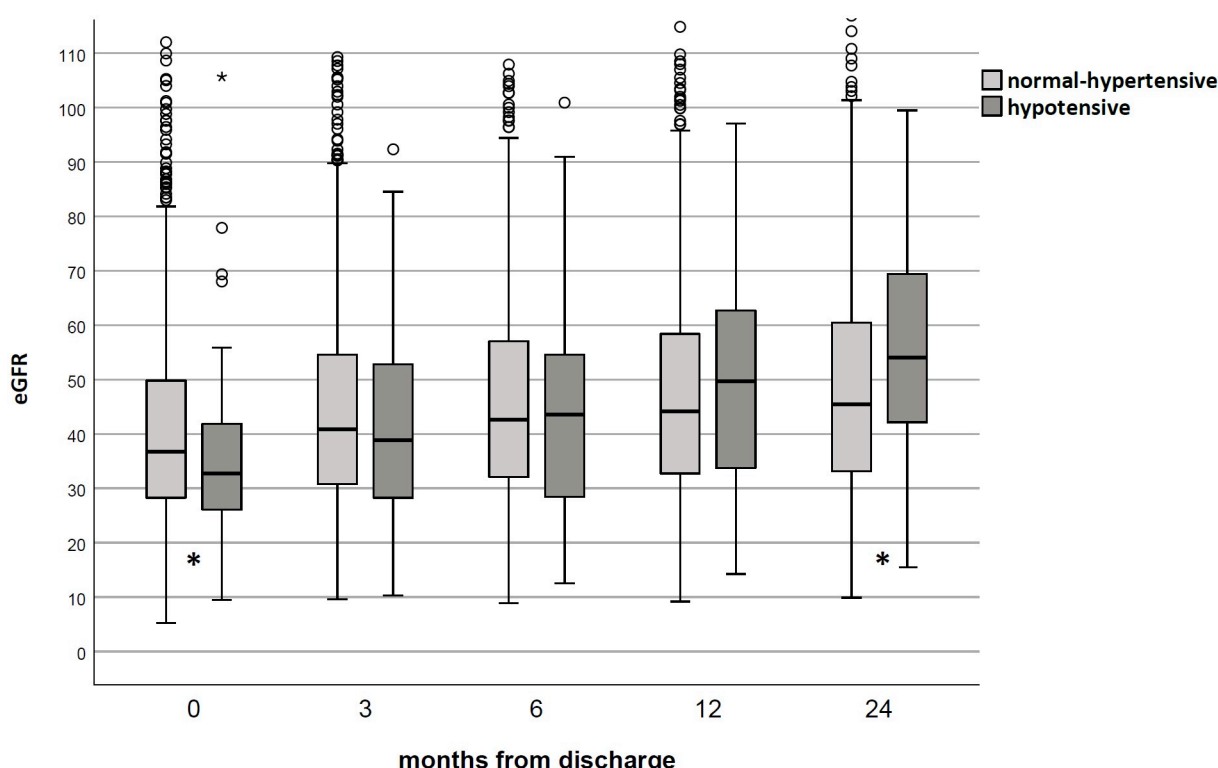

**Fig 5. Renal function in studied population according to pre-existing blood pressure status.** No difference was noted between the hypotensive and normal-hypertensive groups during the follow-up (p = N.S. at any time-point).

incidence was significantly higher in patients with pre-existing hypotension [(66.7% vs. 22.2% in Normal-hypertensive KT; p = 0.02, OR 7.0 (1.6–30.8)] without differences in recipient age, CIT, graft, or patient survival.

Considering perioperative data (available in 917/1127, S1 File, sheet 2), we found a concordance between "historical" hypotension and an MBP<80 mmHg immediately before, during (at reperfusion), or after the surgery (K = 0.616); however, pre-existing hypotension showed a better associated with DGF [OR for perioperative hypotension 2.8 (1.9–4.1)].

Additionally, patients in the hypotension group also had an increased risk of receiving a perioperative fluid administration > 3000 mL (37.9% vs. 13.9% in normal-hypertensive; p<0.001).

## Discussion

Renal function is intrinsically related to hemodynamic status. Autoregulation allows kidneys to maintain normal blood flow and glomerular filtration rate between a mean systemic arterial pressure from 80 mmHg to 160 mmHg [16]. However, in the first post-transplant period, some critical stress factors (i.e., ischemia-reperfusion injury, exposure to vasoconstrictors such as calcineurin inhibitors) may impair this system [4, 17], so the effect of hypotension could be further aggravated with a significant reduction in renal blood flow.

However, few studies have specifically investigated the possible consequences of recipient hypotension in graft and patient outcomes, often limiting their considerations to perioperative management or short-term post-transplant period.

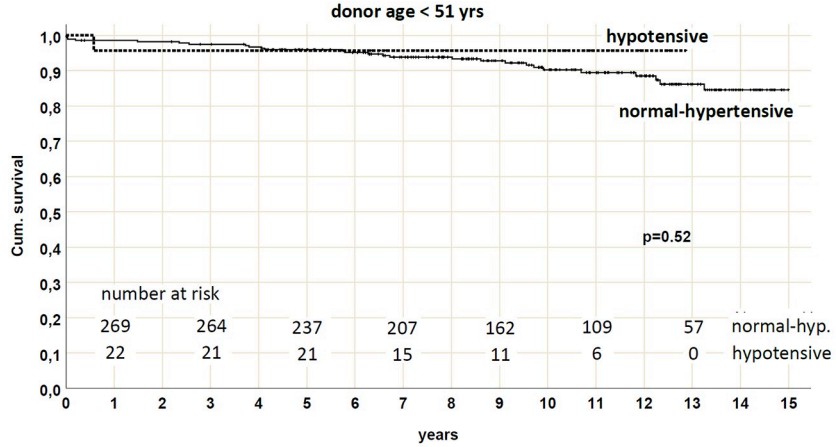

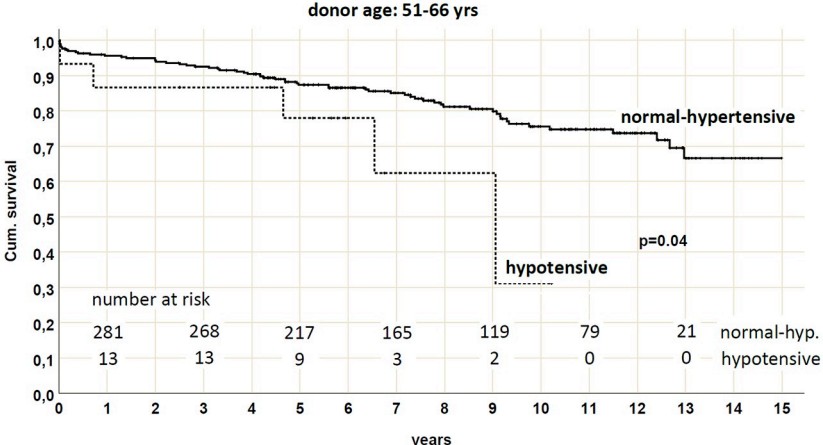

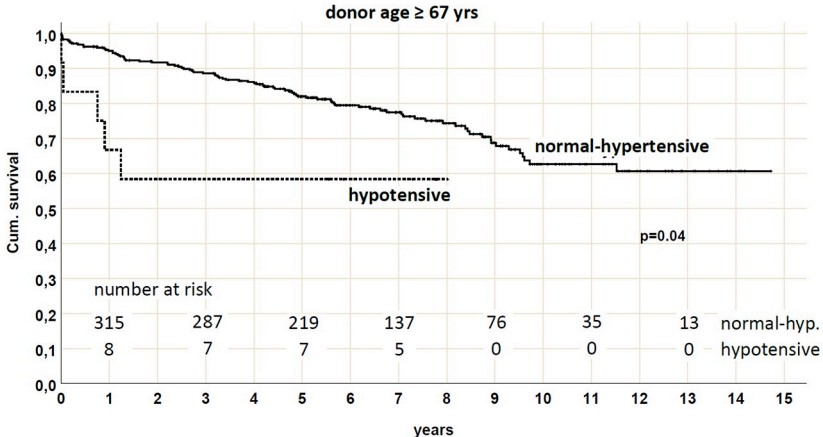

**Fig 6. Death-censored graft survival by pre-existing blood pressure status and stratified by donor age.** Death-censored graft survival was reduced in patients with pre-existing hypotension who received a kidney from a donor between 51 and 66 yrs (p = 0.04) or ≥ 67 yrs (p = 0.04).

**Table 6. Multivariate Cox regression analysis.**

|  | P | H.R. | 95% C.I. per EXP(B) | |
|---|---|---|---|---|
|  |  |  | Inferior | Superior |
| **MBP<80 mmHg** | 0.146 | 1.593 | 0.851 | 2.984 |
| **DGF** | <0.001 | 2.177 | 1.599 | 2.963 |
| **Donor age** | <0.01 | 1.037 | 1.026 | 1.049 |

MBP: Mean blood pressure; DGF: delayed graft function

In our study, we found that pre-existing chronic hypotension (defined as an MBP<80 mmHg assessed during a three months pre-transplant period) is a crucial detrimental factor because a) it significantly and independently correlates to DGF, also in paired kidney analysis among KTs with discordant blood pressure status (hypotensive vs. normal-hypertensive); b) in KTs from elderly donors it may contribute to a low death-censored graft survival due to hypotension-related DGF; c) it is associated with documented intraoperative hypotensive episodes and perioperative fluid administration > 3000 mL with a better correlation to DGF than perioperative values.

Even though DGF is broadly considered a harmful condition with a significant negative impact on graft and patient survival [18–22], its relationship with hypotension is debated.

In 2000, Boom [1] reported that recipients with MBP below 100 mmHg before transplantation had a doubled risk of DGF. However, the studied population (CIT 29±7 h, donor age 37±14 yrs, recipient age 46±13 yrs) was quite different from standard real-life settings.

Ozdemir et al. [2] associated a systolic blood pressure <120 mmHg to an increased risk of DGF in living-related renal allograft recipients: these data again involved a limited population (KTs from living donors) with a younger age than typically observed in current practice.

Our data are closely related to those reported by Gingell-Littlejohn et al. [11] and Kaufmann et al. [9]. Both studies identified a correlation between intraoperative MBP and DGF occurrence, despite targets and cut-offs varied between studies [any recorded MBP<70 mmHg and an MBP <80 mm Hg at the time of reperfusion, respectively]. As for Webber et al. [4], who adopted our definition observing the highest risk of PNF for patients with MBP <80 mmHg, the high OR identified by multivariate analysis and confirmed in both paired kidneys and perioperative data analyses suggest that, besides being only related to intraoperative hypotension, chronic pre-existing blood pressure status may better describe a high-risk profile of KTs who may develop DGF.

Additionally, to the best of our knowledge, our study is the first to deeply evaluate long-term graft and patient survival stratifying population according to pre-existing blood pressure status and with the absence of confounding factors such as extensive percentage of living donation or donors after cardiac deaths.

We did not find an association between hypotension and graft or patient survival on the overall population and paired kidneys. However, sorting out the population by donor age, hypotensive patients who received KT from a donor >50 yrs had a reduced death-censored graft survival; according to multivariate Cox regression analysis, where only DGF and donor age remain significant, we speculate that DGF-mediated hypotension may particularly exert its negative effect in this frail subset of KTs, identifying a subgroup of patients where targeted blood pressure monitorization acquires the highest importance.

Our results effectively emphasize the need for rational peri- and postoperative fluid management to maintain adequate kidney perfusion without a maximization of cardiac filling; however, as reported in literature and observed in current practice, this target remains a

challenge [15]. No guideline or recommendation is to date available for catecholamine choice/dosage [15, 23] as for intraoperative fluid amount and quality [24–26]. On the other hand, the excessive use of inotropes [27] and fluid infusions [9], as observed in our population, were both associated with adverse outcomes. A "goal-directed therapy," borrowed from critical care experience [28, 29], may represent the optimal approach in this setting, although no studies are currently available in KTs [15].

We acknowledge that our study has some limitations: the retrospective design, the limited number of postoperative data. However, all 1127 patients underwent the same surgical and clinical management and a subsequent strict and continuous monitorization with a precise schedule followed by our team of transplant nephrologists; additionally, the adopted multivariate model and paired kidney analysis reduced the effect of possible confounders.

In conclusion, pre-existing chronic hypotension is the only modifiable condition with a significant and independent association with post-KT DGF occurrence.

Based on our analysis results, we would also highlight the association between hypotension and reduced death-censored graft survival in elderly patients, suggesting that hypotension-mediated DGF may become more critical in the setting of the so-called "suboptimal" donors, stressing the importance of therapeutic strategies implementation in this area.

Allocation and institutional policies should carefully consider patients with pre-existing chronic hypotension primarily when expanded-criteria donors are used, focusing on maximizing patient outcomes through a corresponding system resource implementation.

## Supporting information

**S1 File. The raw data elaborated in this study are included in supporting information [(sheet 1: Overall population; sheet 2: Perioperative MBP] (last access date 03/10/2021).** (XLSX)

## Author Contributions

**Conceptualization:** Caterina Dolla, Alberto Mella, Giacinta Vigilante, Anna Allesina, Roberto Presta, Aldo Verri, Paolo Gontero, Fabio Gobbi, Roberto Balagna, Roberta Giraudi, Luigi Biancone.

**Data curation:** Caterina Dolla, Giacinta Vigilante, Fabrizio Fop, Anna Allesina.

**Formal analysis:** Caterina Dolla, Giacinta Vigilante, Fabrizio Fop.

**Methodology:** Fabrizio Fop.

**Supervision:** Alberto Mella, Luigi Biancone.

**Validation:** Luigi Biancone.

**Writing – original draft:** Caterina Dolla, Giacinta Vigilante.

**Writing – review & editing:** Alberto Mella, Luigi Biancone.

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
