## [Decision Letter · Decision Letter 0]

10 Nov 2020

PONE-D-20-31022

Recipient pre-existing chronic hypotension is associated with delayed graft function and inferior graft survival in kidney transplant from elderly donors

PLOS ONE

Dear Dr. Biancone,

Thank you for submitting your manuscript to PLOS ONE. After careful consideration, we feel that it has merit but does not fully meet PLOS ONE’s publication criteria as it currently stands. Therefore, we invite you to submit a revised version of the manuscript that addresses the points raised during the review process.

Biancone et al highlight an important and underreported topic in kidney transplantation. Three expert reviewers from all relevant MDT backgrounds in kidney transplantation have commented on the MS, and have indicated major revisions would be necessary to make the MS publishable in PlosOne. I agree with their assessment. It would need to be rewritten with a clear focus on the primary aim of the study. One of the important points made re: the design of the study is that the recipient BP is particularly important after the reperfusion and we are missing details on perioperative details and BP management. Description of methods needs to be better, and expansion of discussion around the limitations of methodology in relation to the outcome of the analyses is required. The MS would need to be corrected/rewritten by a native english speaker given considerable spelling/grammar errors.

We look forward to receiving your revised manuscript.

Kind regards,

Frank JMF Dor, M.D., Ph.D., FEBS, FRCS

Academic Editor

PLOS ONE

Journal Requirements:

2.) Please include the date(s) on which you accessed the databases or records to obtain the data used in your study.

3.) We note that your study involved analysis of data relating to tissue/organ transplantation. Please provide the following additional information regarding tissue/organ donors for transplantation cases analyzed in your study.

1. Please provide the full name of the source of the transplanted tissue/organs used in the study (the transplantation center name).

2. Please state in your response letter and ethics statement whether the transplant cases for this study involved any vulnerable populations; for example, tissue/organs from prisoners, subjects with reduced mental capacity due to illness or age, or minors.

- If a vulnerable population was used, please describe the population, justify the decision to use tissue/organ donations from this group, and clearly describe what measures were taken in the informed consent procedure to assure protection of the vulnerable group and avoid coercion.

- If a vulnerable population was not used, please state in your ethics statement, “None of the transplant donors was from a vulnerable population and all donors or next of kin provided written informed consent that was freely given.”

3. In the Methods, please provide detailed information about the procedure by which signed informed consent was obtained from organ/tissue donors or their next of kin. In addition, please provide a blank example of the form used to obtain consent from donors, and an English translation if the original is in a different language.

4. Please indicate whether the donors were previously registered as organ donors.

5. Please discuss whether medical costs were covered or other cash payments were provided to the family of the donor. If so, please specify the value of this support (in local currency and equivalent to U.S. dollars).

4.) Please include captions for your Supporting Information files at the end of your manuscript, and update any in-text citations to match accordingly. Please see our Supporting Information guidelines for more information: http://journals.plos.org/plosone/s/supporting-information.

Reviewers' comments:

Reviewer's Responses to Questions

**Comments to the Author**

1. Is the manuscript technically sound, and do the data support the conclusions?

Reviewer #1: Yes

Reviewer #2: Partly

Reviewer #3: Partly

2. Has the statistical analysis been performed appropriately and rigorously? 

Reviewer #1: Yes

Reviewer #2: No

Reviewer #3: Yes

3. Have the authors made all data underlying the findings in their manuscript fully available?

Reviewer #1: Yes

Reviewer #2: Yes

Reviewer #3: Yes

4. Is the manuscript presented in an intelligible fashion and written in standard English?

Reviewer #1: Yes

Reviewer #2: No

Reviewer #3: No

5. Review Comments to the Author

Reviewer #1: This is an extremely interesting study focusing on a neglected but critical aspect of kidney transplantation. Recipient hypotension is in my personal experience the main factor underlying delayed graft function - even more important in practice than cold ischaemic time - yet little in the way of academic study has focused on this area.

The study - a retrospective analysis of >1000 cases analyses associations between recipient BP (assessed in multiple ways) in the period PRIOR to transplantation. In fact the recipient BP is only relevant AFTER reperfusion and this important point could be highlighted in the introduction more prominently. There is however likely to be a high correlation with intra-operative and post-operative blood pressure immediately post transplant and the salient point of the article holds true despite this.

The authors use MAP<80mmHg as the definition of hypotension. That is reasonable based on consensus however this issue has been specifically studied in the perioperative context and ROC curve analysis indicates perioperative BP of 75mmHg or less is the threshold below which DGF becomes significantly more likely (Gingell-Littlejohn et al. Transplant Proc . Jan-Feb 2013;45(1):46-50. doi: 10.1016/j.transproceed.2012.03.058. Epub 2012 Sep 6.) This study is relevant to the article and should be quoted.

The further analyses using paired kidneys and propensity matching add strength to the correlation described. The discussion does not adequately capture the importance of the point made nor the obvious ways in which addressing the issue of recipient hypotension might be attempted. The discussion should consider approaches like goal-directed therapy, inotropes, pressors, specific renal artery flow measurement etc. all of which might be logical approaches to minimise the DGF risk in a hypotensive recipient.

Some further specific points:

The article briefly acknowledges the limitations caused by potential covariance (P19 line 313). This needs expansion. Specifically, the length of time spent on dialysis is a key variable which I am certain would have been available to the authors with a little further work. Specifically this would be an ideal parameter to have applied in the propensity matching in order to address the potential confounding effect. It is well known that chronic HD is associated with hypotension and the absence of this parameter undermines the MVA. This issue should also feature in the discussion as it is unclear how modifiable such a state is in the short period between notification and kidney implantation. If this effect is the main origin of recipient hypotension, strategies to address that in chronic dialysis patients - and their obvious difficulties - should be discussed in the manuscript.

The manuscript appropriately ends on highlighting the danger inherent in allocating older donor kidneys to the hypotensive group.

A further important point worth making relates to the context of recent allocation changes - such as the D4/R4 combination favoured by the new UK scheme since Sept. 2019). Based on the paper's analysis it can be more specifically stated that such transplants are likely to have more DGF, longer stays, more post transplant biopsies and require more resource per case. It is therefore reasonable to conclude that s ystems should provide additional resources to address these inevitable issues BEFORE committing overstretched units to repeatedly undertake such transplants by through allocation policy change.

Table s 2 and 8 would be better presented as graphs

Table 5: several P values appear nonsensical

Reviewer #2: This paper is single center study into association of recipient hypotension with delayed graft outcome and graft survival after kidney transplantation. The major outcome is that recipient hypotension is associated with the risk for DCG and the risk for death censored graft loss in the case of older donors.

While the role of recipient hypotension in kidney transplant outcome merits additional research I feel that the current version of this paper does not add much current knowledge in the field and have a number of comments.

1. The paper does not seem to have a clear focus. While the introduction states that the study intends to analyze the role of recipient hypotension in kidney transplantation the first section of the paper focuses on risk factors for DGF. This section does not add much. I would suggest addressing the role of hypotension and graft outcome directly without the largely confirmatory section on risk factors for DGF.

2. The methods section does not adequately describe the multivariable analysis. A very restricted number of variables is entered in the presented model, although a couple more variables could be included based on the results of the univariable analysis and clinical reasoning (e.g dialysis type). Instead of a thorough multivariable analysis propensity score matching is included. As a bias for receiving a specific treatment is not an issue in this study, I do not understand the rationale for propensity score matching. A more thorough multivariable analysis with the possible addition of a mediation analysis would seem preferable to me.

3. The visual quality of the Kaplan Meier curves is poor. I would also appreciate the addition of numbers at risk in view.

4. I do not completely understand how the number in table 3 add up for dialysis type. If 31.1% of the patient without DGF were on PD, how kan 75.6% of the patients be on HD? This sum is 106.7%.

5. I would leave out the serum creatinine determinations in table 2 and 8. This does not add much next to the eGFR.

6. The text contains a large number of textual errors and needs editing, preferably by a native speaker.

Reviewer #3: This is a retrospective observational study that offers an interesting perspective on a topic that has not been well studied, and may have implications for the perioperative management of kidney transplant patients.

Study design: it is unclear why the authors have chosen MAP 80 as their cut off and, although they correctly assert in the introduction that there is no consensus, it would have been useful to understand their rationale. In the perioperative literature, must studies target MAP 70-90 as a trade off between ensuring adequate renal blood flow, and minimising the vasoconstrictive effects of the agents used to raise the blood pressure intra-operatively. It is likely that were there to be an association between hypotension and graft outcome, this would be as a result of the management in the perioperative period. It would have been useful, therefore, to include data and analysis of the BP immediately before, during and after the surgery to understand if this altered outcome in any way.

Also, what else changed during the period being studied? How was the recipient managed and what factors were taken into consideration? In particular, did the patient's pre-existing blood pressure alter the MAP target perioperatively?

Results: Table 1 - the number of hypertensive patients in line 3 (900) does not match the number in the study design (935). In light of the lack of discussion around MAP targets, inevitably I am left wondering what the results of the study would have been had the MAP cut off been different.

Discussion: the authors have not paid enough attention to a discussion around the effect of the recipient's chronic blood pressure on their management perioperatively, and in particular how the blood pressure and other variables of cardiac output were managed. We know that flow is particularly important in the denervated transplanted kidney - how was this monitored and managed? Could the intra-operative management of the hypotensive patients explain the poorer outcomes?

Overall, this paper raises some pertinent questions; however, the study design at present limits its relevance to clinical practice given the fact that data around the perioperative management of the patients has not been studied.

6. PLOS authors have the option to publish the peer review history of their article (what does this mean?). If published, this will include your full peer review and any attached files.

Reviewer #1: **Yes: **Marc J Clancy

Reviewer #2: No

Reviewer #3: No

---

## [Author Response · Author response to Decision Letter 0]

22 Jan 2021

Academic editor

Biancone et al highlight an important and underreported topic in kidney transplantation. Three expert reviewers from all relevant MDT backgrounds in kidney transplantation have commented on the MS, and have indicated major revisions would be necessary to make the MS publishable in PlosOne. I agree with their assessment. It would need to be rewritten with a clear focus on the primary aim of the study. One of the important points made re: the design of the study is that the recipient BP is particularly important after the reperfusion and we are missing details on perioperative details and BP management. Description of methods needs to be better, and expansion of discussion around the limitations of methodology in relation to the outcome of the analyses is required. The MS would need to be corrected/rewritten by a native english speaker given considerable spelling/grammar errors.

Response. We thank the Academic Editor for his consideration of our paper. According to Academic Editor and reviewers' suggestion, we performed an in-depth revision reconsidering our manuscript with a clear focus on its primary aim (the role of pre-existing chronic hypotension in DGF occurrence and its possible effects on long-term outcomes); additionally, we expanded and rewrote methods, results, and discussion including all findings of available perioperative data and blood pressure management. At least, as suggested, the English language has been evaluated by a native English speaker. We hope that all reviewers' questions are now adequately addressed, and the new version of the manuscript will be suitable for publication in Plos One.

Reviewer #1: This is an extremely interesting study focusing on a neglected but critical aspect of kidney transplantation. Recipient hypotension is in my personal experience the main factor underlying delayed graft function - even more important in practice than cold ischaemic time - yet little in the way of academic study has focused on this area.

The study - a retrospective analysis of >1000 cases analyses associations between recipient BP (assessed in multiple ways) in the period PRIOR to transplantation. In fact the recipient BP is only relevant AFTER reperfusion and this important point could be highlighted in the introduction more prominently. There is however likely to be a high correlation with intra-operative and post-operative blood pressure immediately post transplant and the salient point of the article holds true despite this.

The authors use MAP<80mmHg as the definition of hypotension. That is reasonable based on consensus however this issue has been specifically studied in the perioperative context and ROC curve analysis indicates perioperative BP of 75mmHg or less is the threshold below which DGF becomes significantly more likely (Gingell-Littlejohn et al. Transplant Proc . Jan-Feb 2013;45(1):46-50. doi: 10.1016/j.transproceed.2012.03.058. Epub 2012 Sep 6.) This study is relevant to the article and should be quoted.

The further analyses using paired kidneys and propensity matching add strength to the correlation described. The discussion does not adequately capture the importance of the point made nor the obvious ways in which addressing the issue of recipient hypotension might be attempted. The discussion should consider approaches like goal-directed therapy, inotropes, pressors, specific renal artery flow measurement etc. all of which might be logical approaches to minimise the DGF risk in a hypotensive recipient.

Some further specific points:

The article briefly acknowledges the limitations caused by potential covariance (P19 line 313). This needs expansion. Specifically, the length of time spent on dialysis is a key variable which I am certain would have been available to the authors with a little further work. Specifically this would be an ideal parameter to have applied in the propensity matching in order to address the potential confounding effect. It is well known that chronic HD is associated with hypotension and the absence of this parameter undermines the MVA. This issue should also feature in the discussion as it is unclear how modifiable such a state is in the short period between notification and kidney implantation. If this effect is the main origin of recipient hypotension, strategies to address that in chronic dialysis patients - and their obvious difficulties - should be discussed in the manuscript.

The manuscript appropriately ends on highlighting the danger inherent in allocating older donor kidneys to the hypotensive group.

A further important point worth making relates to the context of recent allocation changes - such as the D4/R4 combination favoured by the new UK scheme since Sept. 2019). Based on the paper's analysis it can be more specifically stated that such transplants are likely to have more DGF, longer stays, more post transplant biopsies and require more resource per case. It is therefore reasonable to conclude that s ystems should provide additional resources to address these inevitable issues BEFORE committing overstretched units to repeatedly undertake such transplants by through allocation policy change.

Table s 2 and 8 would be better presented as graphs

Table 5: several P values appear nonsensical

Response: We appreciate all Rev#1 comments that allow us to improve our paper's quality. According to his suggestions, we have evaluated all our manuscript, also including the available perioperative data. We have consequently rewritten the introduction, methods, results, and discussion focusing on the importance of perioperative management and commenting on the study results by Gingell-Littlejohn et al. (now Ref#11) in introduction and discussion.

In the discussion, we have also included an entire section about the need for a rational perioperative approach to minimize hypotension and its potential detrimental role for transplant outcome, stressing the importance of goal-direct therapies (Page 16-17, Line 292-300).

According to Rev#2 comments, we also implemented our model eliminating the propensity score and modifying the multivariate analysis, even including the length of time spent on dialysis, which is significant in both univariate and multivariate analysis (Table 2); in our opinion, the addition of this substantial variable (for which the thank Rev#1) stress the importance and the generalizability of our results.

As requested, Tables 2 and 8 were presented as graphs (now Figure 2 and 5), and Table 5 (now Table 4) was rewritten, including appropriate statistical analysis.

As also appropriately suggested, we stated in the last part of the discussion that allocation and institutional policies should carefully consider patients with pre-existing chronic hypotension, primarily when expanded-criteria donors are used, focusing on maximizing patient outcomes with a corresponding implementation of system resources (Page 17, Line 312-314).

 

Reviewer #2: This paper is single center study into association of recipient hypotension with delayed graft outcome and graft survival after kidney transplantation. The major outcome is that recipient hypotension is associated with the risk for DCG and the risk for death censored graft loss in the case of older donors.

While the role of recipient hypotension in kidney transplant outcome merits additional research I feel that the current version of this paper does not add much current knowledge in the field and have a number of comments.

1. The paper does not seem to have a clear focus. While the introduction states that the study intends to analyze the role of recipient hypotension in kidney transplantation the first section of the paper focuses on risk factors for DGF. This section does not add much. I would suggest addressing the role of hypotension and graft outcome directly without the largely confirmatory section on risk factors for DGF.

Response: We appreciate Rev#2 considerations because they allow us to refine our analysis. As reported in the Academic editor and Rev#1 response, we modified the introduction (and the whole manuscript), primarily focusing on our study's aim (the role of pre-existing chronic hypotension in DGF occurrence and its possible effects on long-term outcomes) and after that discussing all the implications of our findings.

2. The methods section does not adequately describe the multivariable analysis. A very restricted number of variables is entered in the presented model, although a couple more variables could be included based on the results of the univariable analysis and clinical reasoning (e.g dialysis type). Instead of a thorough multivariable analysis propensity score matching is included. As a bias for receiving a specific treatment is not an issue in this study, I do not understand the rationale for propensity score matching. A more thorough multivariable analysis with the possible addition of a mediation analysis would seem preferable to me.

Response: As reported in Rev#1 comment, we implemented our model eliminating the propensity score and modifying the multivariate analysis, also including, for example, the length of time spent on dialysis, which is significant in univariate and multivariate analysis; in our opinion, the implementation of our research stresses the importance and the generalizability of our results.

3. The visual quality of the Kaplan Meier curves is poor. I would also appreciate the addition of numbers at risk in view.

Response: We modified Figures 1, 4, and 6 according to the requested resolution format, adding the numbers at risk.

4. I do not completely understand how the number in table 3 add up for dialysis type. If 31.1% of the patient without DGF were on PD, how kan 75.6% of the patients be on HD? This sum is 106.7%.

Response: We apologize for this error. As expressed in all comments, we performed an in-depth revaluation of all results, correcting and revisioning all data. In the updated version, table 2 now reported the correct percentages.

5. I would leave out the serum creatinine determinations in table 2 and 8. This does not add much next to the eGFR.

Response: Serum creatinine determinations have been eliminated; as suggested by Rev#1, Tables 2 and 8 are now expressed as graphs (Figures 2 and 5, respectively).

6. The text contains a large number of textual errors and needs editing, preferably by a native speaker.

Response: As also requested by Academic Editor, the manuscript language has been reviewed by a native English speaker.

 

Reviewer #3: This is a retrospective observational study that offers an interesting perspective on a topic that has not been well studied, and may have implications for the perioperative management of kidney transplant patients.

Study design: it is unclear why the authors have chosen MAP 80 as their cut off and, although they correctly assert in the introduction that there is no consensus, it would have been useful to understand their rationale. In the perioperative Literature, must studies target MAP 70-90 as a trade off between ensuring adequate renal blood flow, and minimising the vasoconstrictive effects of the agents used to raise the blood pressure intra-operatively. It is likely that were there to be an association between hypotension and graft outcome, this would be as a result of the management in the perioperative period. It would have been useful, therefore, to include data and analysis of the BP immediately before, during and after the surgery to understand if this altered outcome in any way.

Also, what else changed during the period being studied? How was the recipient managed and what factors were taken into consideration? In particular, did the patient's pre-existing blood pressure alter the MAP target perioperatively?

Response: We thank Rev#3 for his/her comment. As expressed above, we have evaluated all our paper, including the available perioperative data and rewriting introduction, methods, results, and discussion focusing on the importance of perioperative management. Briefly, as stated in methods, all our patients were similarly managed with a target MBP of 80 mmHg maintained with crystalloids and the association of dopamine in case of persistent sub-target values, and we found a significant correlation between pre-existing and intraoperative hypotension; at the same time, we are aware that the cut-off for hypotension is a matter of debate: however, based on previous experience in literature and the prior demonstration of a correlation between PNF and a pre-transplant MBP<80 mmHg in Webber et al. (Ref#4), we decided to maintain this threshold (as now better expressed in both introduction and methods). The new implementation of multivariate analysis with the demonstration of a higher correlation between pre-existing hypotension and DGF suggests, in our opinion, the importance of the analyzed variable and the need for a consistent and rational perioperative approach to minimize its effect on transplant outcome.

All these considerations are now included in the manuscript, focusing on goal-direct therapies' importance (Page 16-17, Line 292-300).

Results: Table 1 - the number of hypertensive patients in line 3 (900) does not match the number in the study design (935). In light of the lack of discussion around MAP targets, inevitably I am left wondering what the results of the study would have been had the MAP cut off been different.

Response: As for Rev#2, we apologize for this error. As expressed in all comments, we performed an in-depth revaluation of all results, correcting and revisioning all data. Table 1 now included the correct numbers and percentages.

Discussion: the authors have not paid enough attention to a discussion around the effect of the recipient's chronic blood pressure on their management perioperatively, and in particular how the blood pressure and other variables of cardiac output were managed. We know that flow is particularly important in the denervated transplanted kidney - how was this monitored and managed? Could the intra-operative management of the hypotensive patients explain the poorer outcomes?

Overall, this paper raises some pertinent questions; however, the study design at present limits its relevance to clinical practice given the fact that data around the perioperative management of the patients has not been studied.

Response: As mentioned above, we now stated in methods that all our patients were similarly managed with a target MBP of 80 mmHg maintained with crystalloids infusions and dopamine association in case of persistent sub-target values. The analysis of available intraoperative data (before, during – at reperfusion – and after the surgery) identifies a close correlation between "historical" and intraoperative hypotension and an increased administered volume (>3000 mL) in hypotensive patients. We agree with Rev#3 that the number of perioperative data is a limitation of our study (as for most of the available Literature papers about this topic); however, the implementation of multivariate analysis with the demonstration of a higher correlation between pre-existing hypotension and DGF suggest, in our opinion, the importance of the analyzed variable and the need of a consistent and rational perioperative approach to minimize its effect on transplant outcome.

---

## [Decision Letter · Decision Letter 1]

11 Feb 2021

PONE-D-20-31022R1

Recipient pre-existing chronic hypotension is associated with delayed graft function and inferior graft survival in kidney transplantation from elderly donors

PLOS ONE

Dear Dr. Biancone,

Thank you for submitting your manuscript to PLOS ONE. After careful consideration, we feel that it has merit but does not fully meet PLOS ONE’s publication criteria as it currently stands. Therefore, we invite you to submit a revised version of the manuscript that addresses the points raised during the review process.

ACADEMIC EDITOR:

Many thanks for making the revisions as per the reviewers' recommendations. They certainly have improved the quality of the MS, but the reviewers still have a few outstanding points that need to be addressed, as i think they are important. They are specified in the comments below. I would encourage you to thoroughly go through them and make the appropriate revisions. I think the MS should be acceptable for publication after that, but will depend on the quality of the revisions and will undergo thorough re-review especially in the light of the outstanding concerns.

We look forward to receiving your revised manuscript.

Kind regards,

Frank JMF Dor, M.D., Ph.D., FEBS, FRCS

Academic Editor

PLOS ONE

Reviewers' comments:

Reviewer's Responses to Questions

**Comments to the Author**

1. If the authors have adequately addressed your comments raised in a previous round of review and you feel that this manuscript is now acceptable for publication, you may indicate that here to bypass the “Comments to the Author” section, enter your conflict of interest statement in the “Confidential to Editor” section, and submit your "Accept" recommendation.

Reviewer #1: All comments have been addressed

Reviewer #2: All comments have been addressed

Reviewer #3: (No Response)

2. Is the manuscript technically sound, and do the data support the conclusions?

Reviewer #1: Yes

Reviewer #2: Yes

Reviewer #3: Partly

3. Has the statistical analysis been performed appropriately and rigorously? 

Reviewer #1: Yes

Reviewer #2: Yes

Reviewer #3: Yes

4. Have the authors made all data underlying the findings in their manuscript fully available?

Reviewer #1: Yes

Reviewer #2: Yes

Reviewer #3: Yes

5. Is the manuscript presented in an intelligible fashion and written in standard English?

Reviewer #1: Yes

Reviewer #2: Yes

Reviewer #3: No

6. Review Comments to the Author

Reviewer #1: The authors appear to have carefully considered the extensive review comments. the revised version is much clearer and of value to the journal's readership in highlighting the important consequences of recipient hypotension for kidney transplantation. the manuscript highlights the importance of considering this factor in both policy decisions around allocation and actual management of individual patients. the manuscript makes appropriate suggestions regarding further studies in the area in line with initial reviews.

Reviewer #2: After the thorough revision the paper is much more focused and has clearly improved. The writing is also clearly improved . I do have one question about the multivariate analysis as shown in table 5. The authors state that the included the significant predictors for DGF as shown in table 2. However the highly significant predictors type of dialysis and donor hypertension are left out. I believe this is not explained in the manuscript. Especially type of dialysis seams an important determinant to ad in this analysis.

Reviewer #3: Thank you for your revisions and I acknowledge the significant amount of work that the authors have put into the manuscript. However, I have two significant concerns:

1. There remain some significant English grammatical and spelling errors particularly in the introduction, methods and discussion. These would need to be resolved prior to the publication;

2. My previous comments regarding the management in the perioperative period. Having looked at the supplementary data provided, this is insufficient to provide analysis: data are only provided for 89 of the 1127 receipients and it is difficult to draw any meaningful conclusions from the data presented. The outstanding issue of how those patients with pre-existing hypotension were managed in the perioperative still needs to be resolved.

7. PLOS authors have the option to publish the peer review history of their article (what does this mean?). If published, this will include your full peer review and any attached files.

Reviewer #1: No

Reviewer #2: No

Reviewer #3: **Yes: **Dr Marc Wittenberg BSc(Hons), MBChB, FRCA

---

## [Author Response · Author response to Decision Letter 1]

11 Mar 2021

Academic editor

Many thanks for making the revisions as per the reviewers’ recommendations. They certainly have improved the quality of the MS, but the reviewers still have a few outstanding points that need to be addressed, as i think they are important. They are specified in the comments below. I would encourage you to thoroughly go through them and make the appropriate revisions. I think the MS should be acceptable for publication after that, but will depend on the quality of the revisions and will undergo thorough re-review especially in the light of the outstanding concerns.

Response. We thank the Academic Editor for his comments. According to reviewers’ suggestion, we re-checked patients schedule for perioperative data (which are now available in 917/1127); as reported below, we found a high concordance between “historical” hypotension and an MBP<80 mmHg immediately before, during (at reperfusion), or after the surgery (K=0.616); however, pre-existing hypotension showed a better association with DGF. As now better expressed in results and discussion sections, our findings suggest that, besides being only related to intraoperative hypotension, chronic pre-existing blood pressure status may better describe a high-risk profile of KTs who may develop DGF.

As also indicated by Rev#2, we implemented the multivariate analysis including dialysis type and donor hypertension and confirming that patients in the hypotensive group had the higher risk of experiencing DGF (OR, 5.4; 95% confidence interval [CI], 2.8 to 10.6). As also suggested, the English language has been extensively evaluated.

 

Reviewer #1: The authors appear to have carefully considered the extensive review comments. the revised version is much clearer and of value to the journal’s readership in highlighting the important consequences of recipient hypotension for kidney transplantation. the manuscript highlights the importance of considering this factor in both policy decisions around allocation and actual management of individual patients. the manuscript makes appropriate suggestions regarding further studies in the area in line with initial reviews.

Response: We appreciate Rev#1 consideration for our paper.

Reviewer #2: After the thorough revision the paper is much more focused and has clearly improved. The writing is also clearly improved . I do have one question about the multivariate analysis as shown in table 5. The authors state that the included the significant predictors for DGF as shown in table 2. However the highly significant predictors type of dialysis and donor hypertension are left out. I believe this is not explained in the manuscript. Especially type of dialysis seams an important determinant to ad in this analysis.

Response: We agree with Rev#2 comments, and, according to his/her suggestions, we implemented the multivariate analysis, including dialysis type and donor hypertension. This final elaboration confirms that patients in the hypotensive group had the higher risk of experiencing DGF (OR, 5.4; 95% confidence interval [CI], 2.8 to 10.6).

Reviewer #3: Thank you for your revisions and I acknowledge the significant amount of work that the authors have put into the manuscript. However, I have two significant concerns:

1. There remain some significant English grammatical and spelling errors particularly in the introduction, methods and discussion. These would need to be resolved prior to the publication;

2. My previous comments regarding the management in the perioperative period. Having looked at the supplementary data provided, this is insufficient to provide analysis: data are only provided for 89 of the 1127 receipients and it is difficult to draw any meaningful conclusions from the data presented. The outstanding issue of how those patients with pre-existing hypotension were managed in the perioperative still needs to be resolved.

Response: We thank Rev#3 for his comments. According to his suggestions:

1- The English language has been further extensively evaluated

2- We re-checked patients schedule for perioperative data (which are now available in 917/1127); as reported, we found a concordance between “historical” hypotension and an MBP<80 mmHg immediately before, during (at reperfusion), or after the surgery (K=0.616); however, pre-existing hypotension showed a better association with DGF [OR for perioperative hypotension 2.8 (1.9-4.1) vs. 5.4 (2.8-10.6)]. As now better expressed in results and discussion sections, our findings suggest that, besides being only related to intraoperative hypotension, chronic pre-existing blood pressure status may better describe a high-risk profile of KTs who may develop DGF. We then stressed in the discussion section the need for a rational peri- and postoperative fluid management to maintain adequate kidney perfusion without a maximization of cardiac filling, considering that a goal-directed therapy may represent the optimal approach in this setting.

---

## [Decision Letter · Decision Letter 2]

22 Mar 2021

Recipient pre-existing chronic hypotension is associated with delayed graft function and inferior graft survival in kidney transplantation from elderly donors

PONE-D-20-31022R2

Dear Dr. Biancone,

We’re pleased to inform you that your manuscript has been judged scientifically suitable for publication and will be formally accepted for publication once it meets all outstanding technical requirements.

Kind regards,

Frank JMF Dor, M.D., Ph.D., FEBS, FRCS

Academic Editor

PLOS ONE

Additional Editor Comments (optional):

Reviewers' comments:

Reviewer's Responses to Questions

**Comments to the Author**

1. If the authors have adequately addressed your comments raised in a previous round of review and you feel that this manuscript is now acceptable for publication, you may indicate that here to bypass the “Comments to the Author” section, enter your conflict of interest statement in the “Confidential to Editor” section, and submit your "Accept" recommendation.

Reviewer #2: All comments have been addressed

Reviewer #3: All comments have been addressed

2. Is the manuscript technically sound, and do the data support the conclusions?

Reviewer #2: Yes

Reviewer #3: Yes

3. Has the statistical analysis been performed appropriately and rigorously? 

Reviewer #2: Yes

Reviewer #3: Yes

4. Have the authors made all data underlying the findings in their manuscript fully available?

Reviewer #2: Yes

Reviewer #3: Yes

5. Is the manuscript presented in an intelligible fashion and written in standard English?

Reviewer #2: Yes

Reviewer #3: Yes

6. Review Comments to the Author

Reviewer #2: (No Response)

Reviewer #3: Thank you for addressing my comments. I am satisfied that they have been adequately addressed and happy to recommend that this paper is published on this important topic.

7. PLOS authors have the option to publish the peer review history of their article (what does this mean?). If published, this will include your full peer review and any attached files.

Reviewer #2: No

Reviewer #3: **Yes: **Dr MD Wittenberg BSc(Hons), MBChB, FRCA, RCPathME

---

## [Editor Report · Acceptance letter]

25 Mar 2021

PONE-D-20-31022R2 

Recipient pre-existing chronic hypotension is associated with delayed graft function and inferior graft survival in kidney transplantation from elderly donors 

Dear Dr. Biancone:

I'm pleased to inform you that your manuscript has been deemed suitable for publication in PLOS ONE. Congratulations! Your manuscript is now with our production department. 

Kind regards, 

on behalf of

Dr. Frank JMF Dor 

Academic Editor

PLOS ONE